# External Morphology of Larvae of *Belgica antarctica* Jacobs, 1900 (Diptera, Chironomidae) Obtained from Two Locations in Maritime Antarctica

**DOI:** 10.3390/insects12090792

**Published:** 2021-09-03

**Authors:** Paraskeva Michailova, Julia Ilkova, Pavlo A. Kovalenko, Volodymyr A. Gorobchyshyn, Iryna A. Kozeretska, Peter Convey

**Affiliations:** 1Institute of Biodiversity and Ecosystem Research, Bulgarian Academy of Sciences, 1 Tzar Osvoboditel b., 1000 Sofia, Bulgaria; pmichailova@yahoo.com (P.M.); juliailkova@yahoo.com (J.I.); 2State Institution Institute for Evolutionary Ecology, National Academy of Sciences of Ukraine, 37 Lebedeva Str., 03143 Kyiv, Ukraine; medziboz@yahoo.com; 3National Antarctic Scientific Center of Ukraine, 16 Taras Shevchenko b., 01601 Kyiv, Ukraine; iryna.kozeretska@gmail.com; 4British Antarctic Survey, NERC, High Cross, Madingley Road, Cambridge CB3 0ET, UK; pcon@bas.ac.uk; 5Department of Zoology, University of Johannesburg, P.O. Box 524, Auckland Park, Johannesburg 2006, South Africa

**Keywords:** antennae, mouthparts, clypeus, pecten epipharyngis, posterior parapods

## Abstract

**Simple Summary:**

The chironomid midge *Belgica antarctica* Jacobs is endemic to the western Antarctic Peninsula and South Shetland Islands. We provide the first detailed photomicrographic images of the fourth-instar larval head capsule and posterior parapods. We assessed variation in the morphology of larvae from two different collection locations off the coast of the western Antarctic Peninsula and compared it with that available in the literature. A number of differences were identified relating to the size of the larvae, the number of teeth on the mandibles, the number of antennal segments and the length of the antennal blade. Malformations of the mandible and mentum are reported for the first time in this species.

**Abstract:**

The external morphology of the fourth-instar larva of the Antarctic endemic chironomid midge *Belgica antarctica* is described. Larvae were collected from Jougla Point (Wiencke Island) and an un-named island close to Enterprise Island, off the coast of the western Antarctic Peninsula. Light microscopy was used to examine and document photographically the structures of the mouthparts (mandible, mentum, premandible, labrum), antennae, pecten epipharyngis, clypeus, frontal apotome and posterior parapods. Measurements of the mouthparts are presented. The data obtained are compared with that available in the literature. A number of differences were identified relating to the size of the larvae, the number of teeth on the mandibles, the number of antennal segments and the length of the antennal blade. Malformations of the mandible and mentum are reported for the first time in this species. Features of larvae of taxonomic value that can be used to determine the species in larval stages are presented. These are of utility in using the larvae to reveal relationships with other species. Larvae are also important in ecological and genotoxicological studies, which require accurate species level identification.

## 1. Introduction

The brachypterous chironomid midge *Belgica antarctica* Jacobs is endemic to the western Antarctic Peninsula and South Shetland Islands. Its biology has been the subject of research by ecologists, physiologists, geneticists and taxonomists [1,2,3,4,5,6,7]. Recent developments in understanding of this insect’s ontogenesis (some stages of which can be studied in vitro) and the large body of studies now available make it an attractive model to study the effects of extreme conditions on biological processes at multiple levels of organization [7].

The adult midge was first described in 1900 based on material collected during the ‘Belgica’ Expedition [8]. The larvae were first described by Rübsaamen (1906) [9], with further information given by Roubard (1907) and Keilin (1912, 1913) [10,11,12]. They were re-described by Torres (1953) (as cited by Usher and Edwards (1984) [13], who stated that the findings were difficult to interpret with the unusual and outdated terminology used). Wirth and Gressitt (1967) published a short description of the larva including a general, but not detailed, illustration [14]. Peckham (1971) illustrated the larval head capsule [15], while Atchley and Hilburn (1979) studied morphological variation in the capsule in relation to environmental conditions, but did not include illustrations [16]. Sugg et al. (1983) studied the four larval instars, providing an illustration of the fourth instar [17]. Usher and Edwards (1984) also provided a detailed, illustrated, description of the fourth-instar larva and drew connections with important taxonomic characters of the subfamily Orthocladiinae [13].

In this study, we provide detailed photomicrographic images of significant elements of the larval morphology, comparing these with information presented in the existing literature. We also assessed variation in the morphology of larvae from two different collection locations off the coast of the western Antarctic Peninsula.

## 2. Materials and Methods

Larvae were collected in February 2020 during the 25th Ukrainian Antarctic expedition. All specimens were preserved in 3:1 ethanol–acetic acid fixative for cytogenetic analysis [18]. The samples were held at −20 °C and transported to Ukraine. Upon arrival, the ethyl ethanol–acetic acid fixative was changed.

We analyzed *B. antarctica* collections obtained from terraces covered by carpets of the moss *Sanionia* sp. from two locations (Figure 1): Jougla Point, Wiencke Island, 500 m south-west of Port Lockroy (03.02.2020; 64.82907° S, 63.48868° W, 32 m a.s.l.) and an un-named island close to Enterprise Island (01.02.2020; 64.55345° S, 61.99451° W, 3 m a.s.l.). 

The external morphology of ten fourth-instar larvae obtained from the un-named island and eleven from Jougla Point (instar determined according to Wülker and Götz (1968) [19]) was examined. Before mounting on slides, individual larvae were placed in 10% KOH for 3–4 h. They were then successively transferred to glacial acetic acid and 96% ethanol for 3 min each. Finally, each specimen was mounted in Euparal (a resin that is used to make reference preparations of different chironomid stages [20]), where the front and back elements of the head capsule were separated. Observations were performed using a Zeiss Axio Scope A1 microscope at magnification of 40× and 100×. Measurements were made at 20× magnification and are given in µm. The magnification indicated in the photographs was obtained by photographing an object micrometer at the same magnification at which the photographs were taken. We used a built-in Zeiss Axio Scope A1 microscope camera. The contrast and size of the photographs have been corrected using Adobe Photoshop. Morphological terminology and abbreviations follow Sæther (1980) [21]. The slides prepared are deposited in the Institute of Biodiversity and Ecosystem Research, Bulgarian Academy of Sciences, Sofia.

## 3. Results

### Morphology of Belgica antarctica Fourth-Instar Larva

Measurement data for individuals from Jougla Point are presented in Table 1 and Figure 2, Figure 3 and Figure 4, and for individuals from the un-named island near Enterprise Island in Table 2 and Figure 5.

**Antenna** (Figure 3a and Figure 5b): four-segmented, lengths of segments 1 and 2 (L1 and L2) given in Table 1 and Table 2. Segment 1 with Ring Organ (RO) located at about mid length and composed of two discs, one of which is indistinct; segment 2 with blade (bl), reaching the fourth segment, sometimes larger than it (Figure 5b); Lauterborn Organ very well developed, appearing as a segment (Figure 3a and Figure 5b). 

**Labrum** (Figure 3d and Figure 4a) with four setae, SI well developed, plumose type (Figure 4a), SII long, simple (Figure 3d), SIII between two SII, smaller than SII (Figure 4a); SIV very small and dark (Figure 3d).

**Fourth-instar larva**: About 7–8 mm in length. All segments darker dorsally and paler ventrally. General coloration: head dark brown in colour, frontal apotome, mentum with dark brown teeth, mandible with brown to dark brown teeth.

**Mandible** (Figure 2a,b and Figure 5a): mandible size data are given in Table 1 and Table 2. Mandible has five dark teeth, under the last of which is a very pale tooth (Figure 2a,b and Figure 5a). Seta subdentalis long and sharpened to the apex, reaching the last posterior dark tooth; seta interna consists of 6–7 strong and conspicuous branches. Some individuals (19%) had only four mandibular teeth (Figure 2a).

**Mentum** (Figure 2c,d and Figure 5a): median tooth large, with central cleft, five lateral teeth sharply pointed and decreasing in size. Width (W) of median tooth 92–96 µm in individuals from Jougla Point (Table 1), 95–104 µm in individuals from un-named island (Table 2). Ventromental plates poorly developed in individuals from Jougla Point (Figure 2d) and well developed in individuals from the un-named island (Figure 5a). The ventromental plates are not well seen in all Jougla Point specimens studied; seta submenti not branched; distance between setae submenti given in Table 1 and Table 2. A high percentage (23.8%) of the studied individuals from both locations showed mal-formation (Figure 2d) affecting the mentum and the range and asymmetry in level of development and number of lateral teeth. Morphological abnormalities affecting the mentum included underdeveloped middle tooth and underdevelopment of the lateral two teeth, which were often fused.

**Premandible** (Figure 3c,d and Figure 5c,d): Each premandible with four visible teeth: two large teeth apically and two small rounded teeth in inner part, not clearly visible in all studied specimens. In some individuals from the un-named island one of the rounded teeth was slightly developed (Figure 5c).

**Clypeus, frontal apotome** shown in Figure 3d, Figure 4b and Figure 5c. Clypeus trapezoidal, with two simple setae S1 and S2; setae 3–5 simple, located in frontal apotome (Figure 4b and Figure 5c); seta S1 located close to the anterior margin; seta S2 inserted medio-laterally (Figure 3d and Figure 5c). S3 close to the anterior margin of frontal apotome, S4 antero-medially, S5 in median area (Figure 4b and Figure 5c). 

**Pecten epipharyngis** (Figure 3d and Figure 5d): consists of three parts. Ungula clearly visible, U-shaped, rounded and sclerotized (Figure 3b,d and Figure 5d). Basal sclerite (BS) (Figure 3b,d) attached to ungula posteriorly, consists of two triangular parts. Two chaetulae visible, ch.l I and ch.l II (Figure 5d).

**Posterior parapods** (Figure 4c,d): anal seta (as) single, about 49 µm long, supra-anal setae (ss) consists of two groups of strong setae, about 109 µm; ventral tubules absent.

## 4. Discussion

Wirth and Gressitt (1967) and Usher and Edwards (1984) provided detailed data on the morphology of *B. antarctica* larvae [13,14]. Our data are compared with these studies in Table 3. Usher and Edwards (1984) noted differences in the length of the head capsule of individuals obtained from two different Antarctic locations, and suggested that these may relate to the sex of the studied larvae or environmental differences [13]. Our data show differences in the fourth-instar larval length both from the two locations studied here and in comparison with the data of Wirth and Gressitt (1967) and Usher and Edwards (1984) [13,14]. The number of teeth on the mandibles also differed between the three studies. In the present study, we also noted that some larvae had one mandible with five teeth and the other four teeth. Warwick (1990), Wise et al. (2001), Al-Shami (2010), and Youbi et al. (2020) also reported this type of malformation in the genus *Chironomus* [22,23,24,25]. The number of branches in seta interna documented in our study overlapped with that reported by Usher and Edwards (1984) [13]. The number of teeth in the mentum did not differ across the three studies, which indicates the adequacy of our analysis [26]. However, we noted further malformations in these teeth, affecting the middle and some lateral teeth. 

Our data indicated that a different number of antennal segments were present compared to the reports of both Wirth and Gressitt (1967) [14] and Usher and Edwards (1984) [13]. Here, we suggest that the very well developed Lauterborn Organ which typifies the subfamily Orthocladiinae [21], located above the second segment, may have been considered to be a separate segment in the previous studies. The blade of the antenna in individuals collected from Jougla Point reached the end of the last segment while, in our specimens from the un-named island and those studied by Wirth and Gressitt (1967) [14] and Usher and Edwards (1984) [13], the blade extended beyond the last segment. Morphological deformities of antennae may be induced by environmental factors as reported by Bhattacharyay et al. (2005) [27] and Reynolds et al. (2002) [28], possibly explaining why our results differ from previously published data, especially given trace element accumulation in terrestrial ecosystems in the Antarctic Peninsula region [29]. 

The malformations affected the number and appearance of teeth of the mentum, the number of teeth of the mandibles and the length of the antennal blade. The deformities observed here may suggest that the larvae are particularly sensitive to environmental contamination [22]. They appear to be capable of responding to very low levels of contaminants, and have the potential to provide an excellent early warning signal for detecting contaminants in the environment. However, the tendency for deformities to arise in response to contamination varies greatly among chironomid taxa [30], as well as with time of the year [30]. Our previous study of genetic variability in chironomid larvae from contaminated areas showed that the polytene chromosomes of the species were much more sensitive to various contaminants than were morphological structures [31]. The detailed morphological characteristics documented here help ensure the exact determination of the species even in larval stages, which can make a valuable contribution to taxonomic, evolutionary and ecological studies.

Differences were observed in the sizes of some of morphological structures assessed in the current study in specimens from the two sampling locations, including in the length of L1 and L2, width of mentum, distance between setae of mentum, and distance from base to the RO of the first antennal segment (see Table 1 and Table 2). These differences might be due to the different larval environments sampled or be related to sexual dimorphism. Adult female *B. antarctica* are larger than males [16], which is very likely to be reflected also in differences in larval size. Our data provide initial evidence for the existence of differentiation in the external morphology of larvae from populations in different parts of the species’ distribution (as suggested by M. Lebouvier and Y. Delettre, cited as pers. comm. by Allegrucci et al. (2012) [3]). Future research on the nuclear and mitochondrial DNA of individuals from different locations will contribute to resolving the genetic basis of any such differentiation.

## 5. Conclusions

Photomicrographs of some body parts of the fourth-instar larvae of the Antarctic endemic chironomid midge *Belgica antarctica* are presented for the first time. We identify differences relating to the size of the larvae, the number of teeth on the mandibles, the number of antennal segments and the length of the antennal blade. Malformations of the mandible and mentum are reported for the first time for this species.

Individual larvae represent the basic units of biological communities. They integrate molecular, cellular, and organ levels of organization into a single entity and form the building blocks for higher levels of organization (population, community, and ecosystem). In the context of assessing pollution using biomarkers (e.g., morphological deformities and genomic alterations), larval stages of Chironomidae have been identified as important bio-monitoring models for eco-toxicological studies.

## Figures and Tables

**Figure 1 insects-12-00792-f001:**
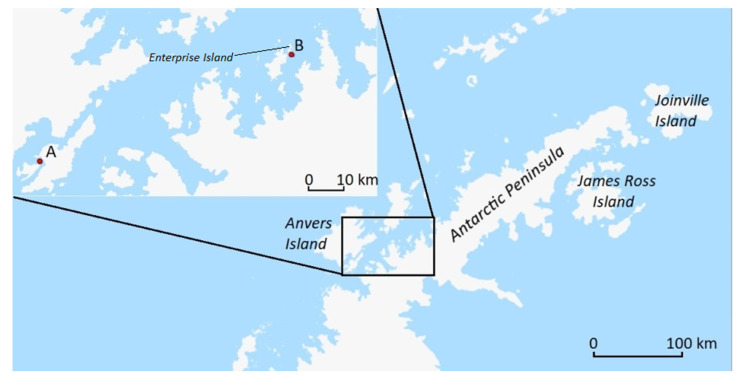
Map showing the locations of sampling sites from which *Belgica antarctica* larvae were obtained: A—Jougla Point, B—un-named island near Enterprise Island.

**Figure 2 insects-12-00792-f002:**
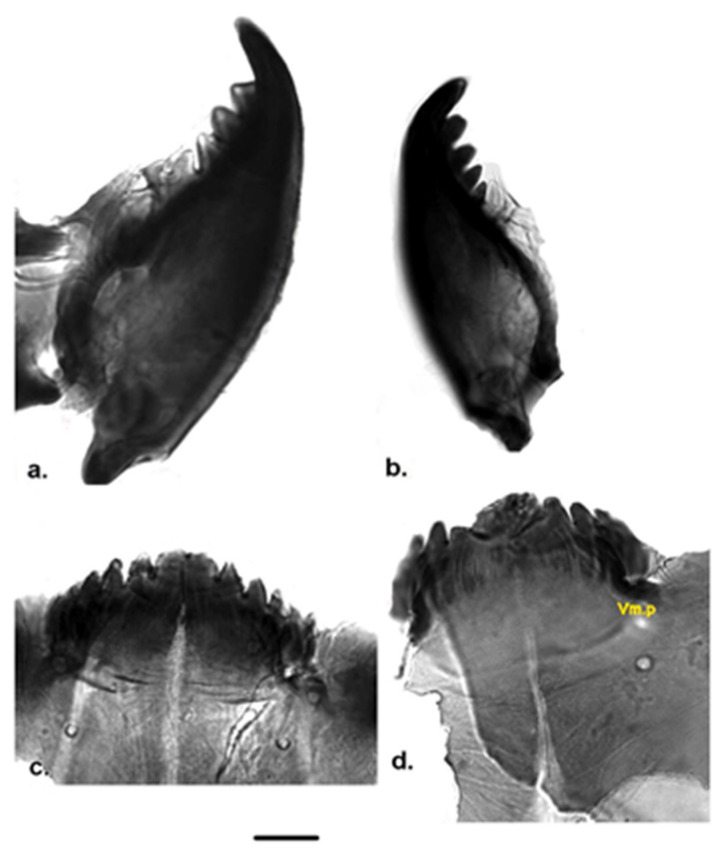
Larva morphology of *Belgica antarctica* (Jougla Point). (**a**) Mandible with four teeth; (**b**) Mandible with five teeth; (**c**) Mentum; (**d**) Mentum with malformation and clearly visible ventromental plate (vm. p). Scale bar 10 µm.

**Figure 3 insects-12-00792-f003:**
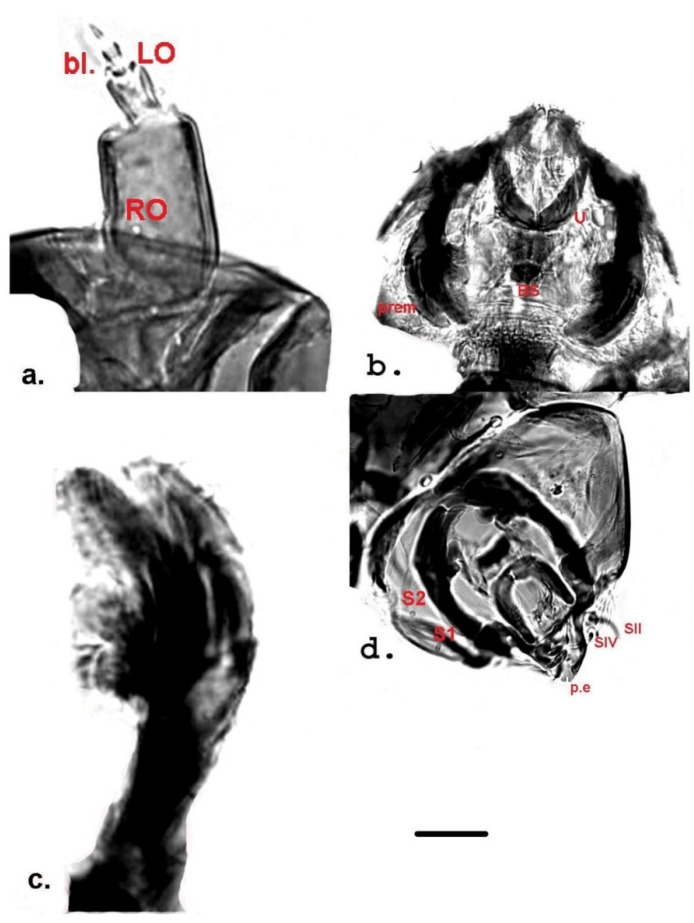
Larval morphology of *Belgica antarctica* (Jougla Point). (**a**) Antenna—blade (bl); Ring Organ (RO); Lauterborn—organ (LO); (**b**) Ungula (U); premandibles (prem.) and basal sclerite (BS); (**c**) Premandible (prem.); (**d**) Pecten epipharyngis (p.e.) and posterior seta (SII) + larger peg of bisensillum (SIV), S1 and S2 of the clypeus; Scale bar 10 µm.

**Figure 4 insects-12-00792-f004:**
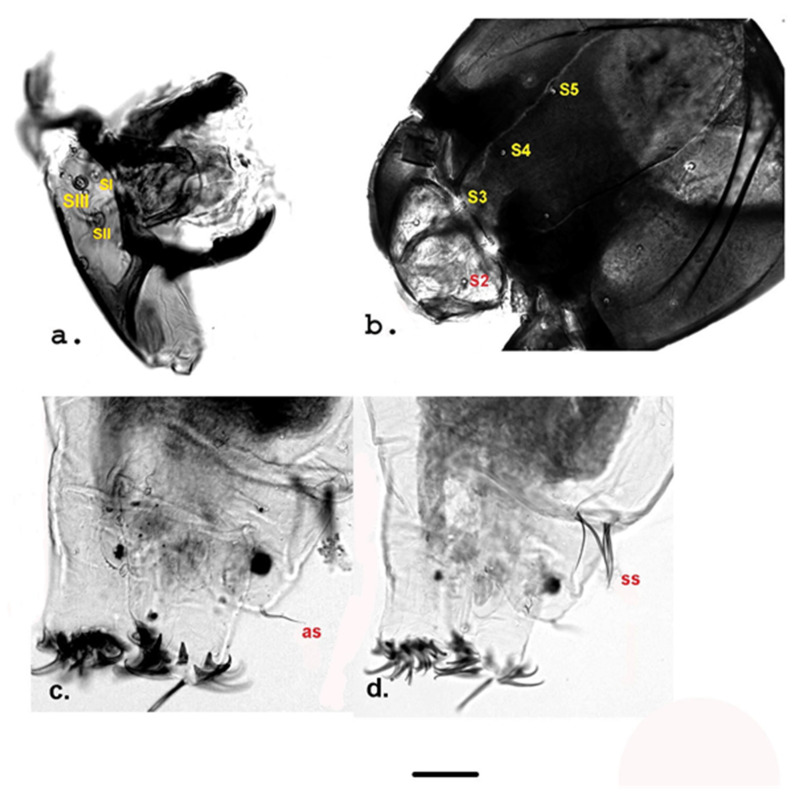
Larval morphology of *Belgica antarctica* (Jougla Point). (**a**) Anterior Seta (SI), posterior seta (SII) and small seta (SIII); (**b**) Clypeus + frontoclypeal area with setae S2, S3, S4, and S5; (**c**) Caudal part of larva with anal setae (as); (**d**) Caudal part of larva with supra-anal setae (ss). Scale bar 10 µm.

**Figure 5 insects-12-00792-f005:**
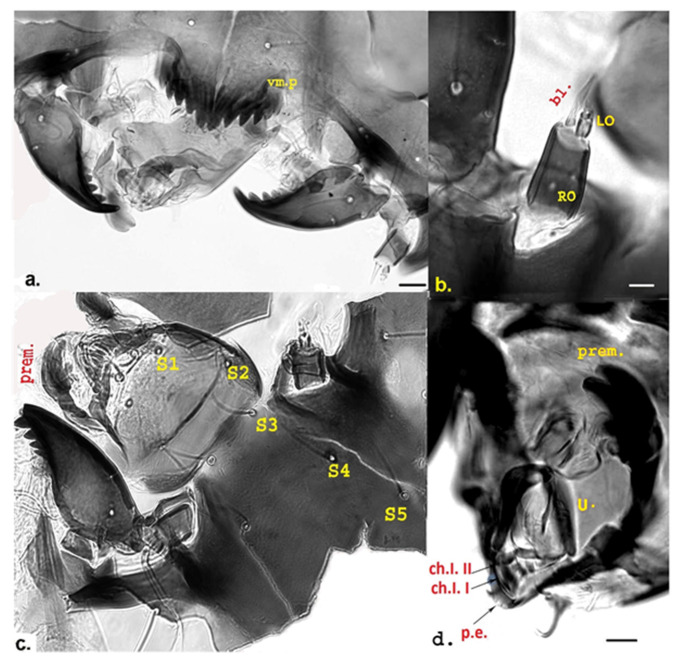
Larva morphology of *Belgica antarctica* (un-named island near Enterprise Island). (**a**) Mentum and mandibles, clearly visible ventromental plate (vm. P.); (**b**) Antenna—blade (bl.); Ring Organ (RO); Lauterborn Organ (LO); (**c**) Clypeus and frontoclypeus with setae (S1–S5), premandible (prem.) and ungula (U.); (**d**) Pecten epipharyngis (p.e); Chaetulae laterales, I-II (ch.1. I-II); premandible (prem). Scale bar 10 µm.

**Table 1 insects-12-00792-t001:** Morphology of 5 of the 11 *Belgica antarctica* fourth-instar larvae collected from Jougla Point. Measurements (in µm) are presented of: Vm.W—width of ventromentum, including the middle tooth + two teeth on both sides, M.W—width of mentum; H. of mand.—height of the mandible; W. of mand—width of mandible; D1—distance between setae of mentum; D2—distance from base to the RO of the first antennal segment; L1 and L2—length of antennal segments 1 and 2.

Vm.W	M.W	H. of Mand.	W. of Mand.	D1	D2	L1	L2
54	92	106	32	65	9	27	8
65	92	107	32	65	9	26	7
60	96	106	32	61	8	27	5
60	92	110	31	68	9	28	8
62	92	110	27	68	9	29	6
60.2 ± 4.0249 *	92.8 ± 1.7888 *	107.8 ± 2.0494 *	30.8 ± 2.1678 *	65.4 ± 2.8809 *	8.8 ± 0.4472 *	27.4 ± 1.402 *	6.8 ± 1.3038 *

* The average values ± SD.

**Table 2 insects-12-00792-t002:** Morphology of 10 *Belgica antarctica* fourth-instar larvae collected from the un-named island near Enterprise Island. Measurements (in µm) are presented of: Vm.W—width of ventromentum, including the middle tooth + two teeth on both sides; M.W.—width of mentum; H. of mand.—height of the mandible; W. of mand.—width of mandible; D1—distance between setae of mentum; D2—distance from base to the RO of the first antennal segment; L1 and L2—length of antennal segments 1 and 2.

Vm.W *n* = 7	M.W *n* = 7	H. of Mand. *n* = 10	W. of Mand. *n* = 10	D1 *n* = 7	D2 *n* = 9	L1 *n* = 9	L2 *n* = 9
65	104	104	41	66	9	22	6
60	100	109	46	65	12	21	6
62	96	111	39	63	10	21	7
62	101	99	44	57	8	21	8
60	98	103	35	67	6	23	6
-	-	107	39	67	7	21	6
-	-	111	43	-	7	19	6
56	96	123	47	-	6	23	5
57	95	105	36	62	-	-	-
-	-	112	43	-	9	20	6
60.2 ± 3.0938 *	98.6 ± 3.2587 *	108.4 ± 6.5862 *	41.3 ± 4.0291 *	63.9 ± 3.5790 *	8.2 ± 1.9861 *	21.2 ± 1.3017 *	6.2 ± 0.8333 *

* The average values ± SD.

**Table 3 insects-12-00792-t003:** Comparative analysis of morphological features of larvae of *Belgica antarctica* assessed in the current study and previously reported by Wirth and Gressitt (1967) [14] and Usher and Edwards (1984) [13].

Larval Feature	Current Study	Wirth and Gressitt 1967	Usher and Edwards 1984
Length	7–8 mm	4.5–5 mm	6–7 mm
Mandible—number of teeth	5; 19% with 4 teeth	5	5
Mentum—number of lateral teeth	5	5	5
Antenna—number of segments	4	5	5
Antennal blade	**Jougla Point**—reached the fourth segment; **un-named island near Enterprise Island**—exceeded the last segment	exceeded the last segment	exceeded the last segment

## Data Availability

Data can be found within the article.

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
