# Peer review of "External Morphology of Larvae of *Belgica antarctica* Jacobs, 1900 (Diptera, Chironomidae) Obtained from Two Locations in Maritime Antarctica"

_insects, 2021, doi:10.3390/insects12090792_

Round 1

Reviewer 1 Report

The manuscript “External morphology of larvae of Belgica antarctica Jacobs, 1900 (Diptera, Chironomidae) obtained from two locations in Maritime Antarctica” provides new details regarding the morphology of this unusual insect and describes how some morphological characters differ from previous descriptions. Belgica antarctica has been extensively studied due to its unusual and extreme habitat and review of its morphology is useful. The study design is largely appropriate and is sufficiently described in the manuscript. The grammar in this manuscript is good and only needs minor revisions before publication. There are also some specific clarifications that should be made. My detailed comments and suggestions are provided below. I recommend that this manuscript be accepted for publication in Insects with minor revisions. 

Specific comments:
The following are specific suggested/ recommended revision to the manuscript. These comments are indexed by the page number and page line provided in the review manuscript. 

Simple summary:
Page 1, line 14: I recommend changing “Chironomid midge” to “The chironomid midge”

Page 1, lines 15-16: I recommend changing “of head capsule of the larval morphology” to “of the morphology of the larval head capsule”

Abstract:
Page 1, line 22: I recommend changing “of the fourth instar larvae” to “the fourth-instar larva”

Page 1, line 32: Can the authors summarize the research in terms of why it is important? How are the results useful?

Introduction:
Page 2, line 38: Can the authors provide a little more detail regarding what is intended by “attracted the attention.” The references are useful, but it would be helpful to have a short (one sentence?) description about what makes this insect so interesting. 

Page 2, line 48: change “larvae” to “larva”

Page 2, line 53: Change “fourth instar” to “fourth-instar.” Check the rest of the manuscript for compound adjectives and hyphenate where appropriate. 

Material and Methods:
Page 2, line 62: I recommend changing “The samples were transported to Ukraine at freezing temperature (-20°C).” to “The samples were and were held at -20 °C and transported to Ukraine.”

Page 2, Figure 1: Would it be possible to label Enterprise Island. This feature is referenced in the locality description, but is not clear where this island is located.

Page 3, line 75: I recommend changing “was included in an Euparal mount” to “was mounted in Euparal.”

Page 3, line 79: Describe the methods for capturing the images used in the figures. What camera was used? It looks like some of the photos have been edited and it may be useful to describe this. 

Results:
Page 3, line 83: Delete extra “s” in “Figuress”.

Page 3, lines 89-90: “H. of mandible – height of the mandible;” should be “H. of mand. – height of the mandible;”

Page 3, line 111-113: The text “In Jougla Point specimens often asymmetrical, not clearly visible in all specimens, laying on the second tooth located laterally;” is not clear. Reword this description to clarify the author’s intent.

Page 3, line 114-116: Can the authors provide more details regarding the deformations? What was the range in the number of lateral teeth? What sort of malformations of the mentum were observed? Could some of this been due to normal wear of the mentum?

Figures 3-5: The labels are difficult to make out in some figures (e.g., Fig. 3b). I recommend improving the visibility of these labels.

Discussion:
Page 7, line 162: Would it be more appropriate to use “and suggested” rather than “considering.” Did Usher and Edwards indicate that the differences were possibly due to the three reasons stated or are these suggested by the authors? 

Page 7. Since in the present paper, all larvae were fourth instars, then instar can be excluded as an effect of variability. The variability between specimens and localities could then be attributed to sex or environmental difference. Correct? It may be worth stating this.

Page 8, lines 166-167: I recommend reword sentence. For example: “The number of teeth on the mandibles also differed between the three studies.”

Page 8, line 167: I recommend changing “In the present study we also noted larvae having one” to “In the present study, we also noted that some larvae had one”

Page 8, line 178: I recommend changing “Our data indicated a different number of antenna segments” to “Our data indicated that a different number of antennal segments”

Page 8, lines 185-189: I recommend creating a separate paragraph to discuss deformities and possible causes. Are toxics accumulating in this area at a level that is likely to cause a high proportion of deformities? What proportion of specimens in this study had deformities and what was the severity of these deformities? Can the authors provide more information here and in the Results section? It is also not clear in the manuscript which structures were observed to have deformities. In the Results section, the only structure where deformities are described is the mentum. However, deformities of the mandible are noted in the Abstract and deformities of the antenna are noted in the Discussion (lines 185-186). Clarify which structures had observed deformities and describe these deformities in the Results section.

Conclusions:
Page 9: Provide more information in this section. In particular, describe why this research important and how can it be used in subsequent research. 

Reviewer 2 Report

The manuscript presents new knowledge and photographic illustration related to variation in the morphology of fourth instar larvae of the chironomid midge Belgica antarctica. This is interesting because of the extreme environment that these larvae live in and the sexual dimorphism observed in the adult midges. The article presents the findings adequately and provides back/white microphotographs of moderate quality to support the descriptions and the discussion, but needs some improvements before publication. Please see list below. Moreover, the work would have been considerably better if genetic data (e.g. DNA barcodes) had been used to confirm that the larvae of the different populations really belong to the same species. Some of the variation described certainly would be sufficient to regard the larvae to belong to different species. 

  1. Please improve the English language in the "Simple summary" and include articles. (e.g. "The chironomid midge ..."). Second sentence should read: "We provide detailed photomicrographic images of the larval head capsule and the larval hind end for the first time"
  2. Methods. Please provide some description of the habitat where the larvae were collected.
  3. Methods. Please provide information on how the larvae were identified to species. How was the instar identified? How does B. antarctica larvae differ from Eretmoptera murphyi larva (or other genera that might be introduced to the Antarctic)
  4. Methods. Do you also use some of the abbreviations in Sæther (1980)? This can ve stated in line 79.
  5. Methods. The magnification listed are the objective magnifications, not those of the microscope since you likely used 10x oculars. Please be precise.
  6. Line 104-105: Please write "seta interna".
  7. Line 105: Please write "mandibular teeth".
  8. Photos: Please improve by removing all the grey dotted fields that are added to the images. These are very disturbing. If you need to "clean" the original images for clarity, do so my removing debris not covering.
  9. Line 160: Plural "larvae"
  10. Line 180: It is "Lauterborn Organ".
  11. Line 200: Please write "... contribute to resolving the genetic basis of any such differentiation."

Reviewer 3 Report

This is a really interesting study of these unique larvae, and over all the paper is great. I had a few minor questions that might need to be addressed in the paper. 

  1. The introduction needs more detail in the speciation of this group. I was constantly wondering if the differences you noted in the specimens were enough to constitute a new species or not. Adding information on that to the intro would have helped with my understanding of what you were explaining.
  2. You mention that the midges were re-described by Torres in 1953, but these were hard to interpret. Was this just because of the mentioned "unusual" terminology? Is it possible to translate, as it were, these terms so they relate to this study? I'm wondering if the information you found in your work is reflected in the Torres study. If it's not possible to clarify this info, then an explanation as to why it's not possible may help, especially since it seems that the Torres paper is a major contribution to this area. 
  3. Given all the differences you have found in these larvae, what are the chances that this is a new species or sub species? You mentioned that DNA and other molecular tests are necessary to really understand what is going on, but given that there are only a few papers describing the larvae, is it possible these are different enough to be named as something new? 

Otherwise, this is a great study, very interesting, and the pictures are fantastic. 
